# Intrathecal Baclofen for Spasticity: Is There an Effect on Bladder Function? Report of Three Cases and Review of the Literature

**DOI:** 10.3390/biomedicines10123266

**Published:** 2022-12-16

**Authors:** Charalampos Konstantinidis, Eleni Moumtzi, Archodia Nicolia, Charalampos Thomas

**Affiliations:** 1Urology & Neuro-Urology Unit, National Rehabilitation Center, 13122 Athens, Greece; 2Physical and Rehabilitation Medicine Department, 414 Military Hospital, 15236 Athens, Greece; 3Urology Department, General Hospital of Corinth, 20100 Corinth, Greece

**Keywords:** traumatic brain injury, baclofen pump, neurogenic bladder

## Abstract

Introduction: traumatic brain injury (TBI) is very often associated with spasticity. Medical interventions may include medications such as baclofen, a Gamma-Aminobutyric Acid (GABA) -receptor agonist of poor lipid solubility. Intrathecal baclofen (ITB) administration is a contemporary treatment option which minimizes adverse effects in contrast with the oral form of the drug. Regarding low urinary tract dysfunction, TBI, as a suprapontine lesion, results in neurogenic detrusor overactivity. Frequency, urgency and urge incontinence are the predominant signs and symptoms of this condition. Our study aims to report the potential changes in bladder function in patients with spasticity, due to TBI, after the implantation of the baclofen pump and the control of spasticity. Material and Methods: We report three cases of TBI whose spasticity responded well to ITB. We evaluated our medical reports regarding bladder function retrospectively, before and after baclofen pump implantation. We compared the data of bladder diaries and urodynamic parameters. Results: Bladder function was improved in all patients. Regarding bladder diaries; the number of incontinence and micturition episodes was decreased and the volume per void was slightly increased. Regarding urodynamic parameters; bladder capacity and reflex volume increased, Pdetmax decreased, PVR was the same and DLPP was slightly decreased. Conclusions: Although the baclofen pump is implanted to treat spasticity, detrusor activity may be also affected. Therefore, patients’ urologic profiles should also be reevaluated after ITB. Further prospective studies are required to investigate the effect of ITB on bladder function in the clinical field and also at the basic science level.

## 1. Introduction

Spasticity presents as a common outcome of an upper motor neuron abnormality. Cerebral and spinal cord lesions may result in spasticity even though the pathophysiological mechanism may be different [1].

Baclofen is a very potent agonist of GABA receptors and is an effective treatment modality for spasticity. The use of baclofen started as an oral agent in the early 1970s [2,3,4]. Unfortunately, its poor lipid solubility demands very high doses in order to pass through the blood–brain barrier and result in effective GABA bounding. This leads to adverse effects such as sedation, excessive weakness, vertigo, and psychological disturbances [5] which make the treatment intolerable. Penn and Kroin [6] first published intrathecal administration of baclofen for spasticity treatment. Intrathecal administration minimizes adverse effects, as the dose needed is 500–1000 times less than the effective oral dose. Nowadays the drug is delivered directly to the cerebrospinal fluid space, through an epidural catheter and the released dose is controlled by an automatic pump which is implanted subcutaneously in the abdomen.

Treating spasticity is very important for patients as their movements become more coordinated and functional, body formation becomes less contractive and more natural, quality of sleep gets better, discomfort or pain decreases and in general quality of life improves [7]. Baclofen leads to muscle relaxation, acting as a GABA receptor agonist which binds to presynaptic and postsynaptic GABAb receptors of the motor neurons in the anterior horns of the spinal cord. Presynaptic binding inhibits the release of neurotransmitters by down-regulating high-voltage calcium channels [8]. Postsynaptic binding inhibits neuronal excitability by activating an inwardly rectifying potassium current [9]. Consequently, the initial signal of muscle contraction is eliminated. Presynaptic action seems to be more important for spasticity reduction [10]. Traumatic brain injury (TBI) is very often associated with spasticity and intrathecal baclofen [ITB] administration is an effective treatment option for this condition [11,12].

Regarding the lower urinary tract function, TBI results predominantly in neurogenic overactivity amongst others (pseudodussynergia and detrusor underactivity). TBI as a suprapontine lesion results in neurogenic detrusor overactivity in the majority of cases, by reducing, central inhibitory mechanisms [13,14]. If there are no additional lesions in the spinal cord, voiding is synergic, as the micturition reflex is controlled by the pontine micturition center. Frequency, urgency and urge incontinence are the predominant signs and symptoms of this condition.

Our study aims to report the potential changes in bladder function in patients with spasticity, due to TBI, after the implantation of a baclofen pump and the control of spasticity.

## 2. Materials and Methods

We report three cases of TBI, whose spasticity responded well to intrathecal baclofen administration. All patients were males, aged 36, 29 and 27 years old and had been injured two, two-and-a-half, and three years respectively before baclofen pump implantation. We evaluated our medical reports referring to bladder function retrospectively, before and after baclofen pump implantation.

Regarding the baclofen pump, it is reported in the medical records of the clinic where they had the pump implanted, that first a temporary catheter was placed into the lumbar region and propelled to the thoracic spinal cord; the bolus test was administered and after the positive reaction of the patients to baclofen they proceeded to the implantation. The ITB dosage was titrated in order to deteriorate the spasticity and allow the normal strength of the patients to manifest. Thus, one patient was titrated up to 200 μg/d, the other up to 300μg/d, and the 3rd one up to 350 μg/d.

Bladder function was evaluated by urodynamics (filling cystometry, pressure-flow study, EMG pelvic floor study, according to the ICS recommendations) [15], before and six months after the pump placement. In all patients, the dose of baclofen administration was regulated and stabilized at least one month before urodynamic investigation. A urodynamic system Dorado KT by Laborie was used. We used a 6 Fr urodynamic 2-way catheter and the bladder was filled with sodium chloride 0.9% at a rate of 20 mL/min at a temperature of 37 °C. Patients were placed in a sitting position to simulate, as much as possible, the conditions of their daily voiding function. The urodynamic parameters taken into account were bladder capacity (BC), reflex volume (RV), post-void residual (PVR), maximum detrusor pressure (Pdetmax) during filling and detrusor leak point pressure (DLPP). All the conditions were the same in both urodynamic investigations, before and after ITB administration. All patients had also the same antimuscarinic medication (solifenacin succinate, 10 mg daily) which they had started by the end of the first post-injury year. Patients have first referred to us two years post-injury for urodynamics examination as they declared episodes of incontinence. They had completed a three-day bladder diary which is a part of our common clinical practice. The bladder diary contained the fluid intake, the number of micturitions, the volume of urine per void and the incontinence episodes.

In our study, we compared the bladder diaries and the urodynamic parameters before and after the baclofen pump implantation.

## 3. Results

Bladder function was improved in all patients. Regarding bladder diaries, incontinence episodes and micturitions were decreased, and the volume per void was slightly increased although the fluid intake was approximately the same. Data of the bladder diaries comparison are summarized in Table 1.

## 4. Discussion

Our initial intention was to express our observation that baclofen pump implantation improved in the long term not only the degree of spasticity for patients with TBI but also their bladder function. We underline that this was accomplished with significantly lower doses of baclofen than orally (usually 15 mg/d up to 80 mg/d divided into doses) and therefore with almost no side effects. The quality of life was improved as there were fewer incontinence problems and the overnight sleep hours were augmented. Maybe it is worth bearing in mind that selecting patients for ITB therapy should depend (apart from the local experience, the severe spasticity lasting at least six months, and having failed to respond to the maximum recommended doses of oral antispasmodic medications) on whether it could result in a concomitant benefit on bladder function in addition to other therapies. It is strongly suggested that patients should be screened for response to the drug before implantation of the delivery pump.

The pathophysiology of neurogenic detrusor overactivity (NDO) depends on several peripheral and central nervous system (CNS) factors. Different CNS disorders are associated with NDO via multifactorial pathological pathways.

Normal micturition in humans and animals depends on afferent signals from the lower urinary tract (LUT), under the control of circuits in the brain and the spinal cord [16,17,18]. Thus, the activity of the smooth muscles (in the detrusor and urethra) and the striated muscle (in the urethral sphincter and pelvic floor) is coordinated. Neurons in the pons are believed to act as “switches” that turn the function from storage to voiding [19]. Consequently, injuries or diseases of CNS in adults may destroy these mechanisms and provoke the emergence of reflex micturition.

Distension of the bladder wall is thought to be the main stimulus for initiation of the bladder reflux. Moreover, the urothelium itself is considered a sensor adjacent to afferent nerves [20]. When the afferent impulses reach the centers in CNS, the periaqueductal grey matter (PAG) is the first region to analyze the impulse and communicates with the pontine tegmentum (PT) for further reaction on behalf of the nervous system [21,22]. In the PT there are two regions involved in micturition control: one dorsomedially located, called M region and corresponding to Barrington’s pontine micturition center (PMC); one laterally located, called L region and corresponding to the pontine urine storage center. These may represent separate and independently functioning systems [23]. Their suprapontine control is not clarified in detail yet [24], although several positron emission tomographies (PET) evince several brain structures [25,26,27]. The circuit described connects via the spinal cord to preganglionic neurons of the lumbar spine and sacral parasympathetic nuclei, thus controlling the striated urethral sphincter.

Lesions emerging on the level below the pontine region may additionally result in detrusor–sphincter dyssynergia [28] while above this region result in several voiding disturbances, such as NDO.

In patients with cortical lesions, voiding is generally coordinated and urine incontinence is attributed to damage of cerebral inhibitory centers, thus uninhibited detrusor contractions [29]. Damage to the anteromedial frontal lobe, descending pathways, and basal ganglia provoked micturition problems in stroke patients, according to Sakakibara et al. [30]. In these patients, NDO was found. Moreover, electromyography revealed uninhibited relaxation of the external sphincter during or before detrusor contractions [29].

Symptoms are usually apparent during the storage phase (frequency, urgency, urge incontinence) and during urodynamics, there is detrusor contraction in low volumes of bladder filling. During the voiding phase, hesitancy and weak urinary stream are often combined with the former ones. In men, the symptomatology could be confused with that of benign prostatic hyperplasia which could also exist.

Different neurotransmitter systems are involved in micturition control and could be potential targets of therapies to control abnormal micturition. Glutamate is probably an excitatory transmitter supraspinally and is also involved in the pathway from the PMC to the preganglionic neuron [31,32]. Antagonists of glutamate receptors in experimentally cerebral infarcted rats deteriorated symptoms of NDO, secondary to the stroke [33,34]. Many CNS functions rely on glutamate which could limit the use of its antagonists for controlling Detrusor Overactivity (DO), as they would affect other functions as well. Other substances, such as GABA, serotonin, dopamine, and noradrenaline, may modulate glutamate’s action on micturition and potentially can be used to intervene for DO.

GABA in particular has been detected in both spinal and supraspinal synapses in mammalian CNS and more precisely in the supraspinal micturition pathway of some of them. Its role is of a rather tonic, inhibitory type [16,19] and may act either on excitatory (diencephalon) or inhibitory (mesencephalon and telencephalon) mechanisms of micturition control. Three types of GABA receptors have been recognized (GABA_A_, GABA_B_, and GABA_C_) and all of them are localized in both the brain and the spinal cord [35,36,37]. Furthermore, GABA transporters have also been detected in the brain, brainstem, and spinal cord [37]. Therefore, not only receptors but also transporters could be targets of therapy. Experiments on conscious and anaesthetized rats have demonstrated inhibition of the micturition reflex by exogenous administration of GABA_A_ receptor agonist muscimol and GABA_B_ receptor agonist baclofen [38,39,40,41]. GABA injected into the pons of cats may decrease or increase bladder capacity, depending on interaction with different brainstem centers of storage or voiding [42]. Baclofen and muscimol administered intra-cerebro-ventricularly in rats with cerebral infarction decreased bladder capacity, whereas they had no effect in normal rats [43]. However, this experiment included low doses of muscimol and baclofen. The same experiment in high doses increased bladder capacity.

Intrathecal administration of the GABA-B agonist baclofen is nowadays generally accepted as a powerful treatment for spasticity of whatever etiology; improvement in mobility and function as well as relief of spastic pain is the most obvious benefit for the patient. The initial technical and methodical problems have been solved and today the procedure is generally assessed as safe [44].

Selecting patients for ITB therapy depends partially on the local experience, but it generally concerns patients with severe spasticity lasting at least six months and having failed to respond to the maximum recommended doses of oral antispasmodic medications either because they are ineffective or because of too many side effects [45]. It is advisable that patient selection for baclofen pump implantation is done in a setting where doctors from different disciplines and several therapeutic modalities can be available if needed. A trial should be performed beforehand in order to confirm that the patient responds well to the drug. The usual dose of ITB, in the end, is stabilized between 90 and 703 μg/day [46].

Complications, while rare, are most often related to the drug delivery catheter [47]. ITB treatment may be cost-effective, primarily due to a reduced need for hospitalizations and treatment of adverse events related to uncontrolled spasticity and secondarily due to improved quality of life. When assessing spasticity, it is of great importance to distinguish whether it is useful for the patient or not. In case of extensive muscle weakness, spasticity may help to maintain an upright position. In other cases, it may be focal, which applies to treatments accordingly (botulinum toxin, peripheral neurotomies etc.). Furthermore, it is important to differentiate spastic patients into those who cannot perform daily activities because they cannot move and those who can, because they set different goals in therapy. Functional index measurement (FIM) scale could help achieve a more complete assessment toward their goals. Regardless of the etiology, ITB treatment is the most effective to date [48].

After ITB implantation, close monitoring is essential to prevent complications and side effects. Either undertreatment or overdose can be life-threatening [49]. The more familiar the clinician is with the ITB system the more concrete the patient’s complaint evaluation. Different treatments at several spinal levels may arise from future research [50].

The current study leads to results that cannot be generalized or applied in everyday practice. The main reasons are the small sample size and the fact that it explains an observation of change in bladder behavior as a positive side effect of therapy used to treat other symptoms. Therefore, prospective studies are required to investigate the effect of ITB on bladder function in the clinical field and at the basic science level.

## 5. Conclusions

ITB is effective, nondestructive, titratable, and reversible. In addition, it is associated with fewer CNS-related side effects than oral baclofen medication. ITB therapy may improve the range of motion, reduce the need for corrective orthopedic surgeries, facilitate movement, reduce the patient’s expenditure of energy, reduce difficulty in nursing interventions, avoid contractures, deteriorate spasticity related pain. It may ameliorate upper and lower extremities’ function or vocal mechanism’s difficulties and therefore improve daily living, and other aspects of self-care. It seems that there is also an effect on bladder function even though our data are not, till now, so strong to prove it, in the field of evidence-based medicine. Physicians conducting urodynamics should be alert and follow such cases in the long term to gather more data on the effect of baclofen pump implantation on lower urinary tract function. Furthermore, other issues arise such as: is it that baclofen affects immediately the detrusor muscle and suppresses its overactivity or is it that baclofen’s action on the urethral sphincter leads to relief, thus to a lower leak point pressure (LPP) and therefore to a lower maximum detrusor pressure (Pdetmax)? Prospective clinical studies have to be designed in order to identify and prove the effect of ITB on bladder function and more studies on basic science are needed to examine the role of GABA on urinary function.

## Figures and Tables

**Table 1 biomedicines-10-03266-t001:** Data of a three day bladder diary before and after ITB.

Data of a Three Day Bladder Diary					
		Fluid Intake (mL) Mean—(Range)	Number of Micturitions Mean—Range	Incontinence Episodes Mean—Range	Volume per Void (mL) Mean—Range
1st patient	Before baclofen	2200—(2000–2400)	14—(13–15)	2—(1–3)	150—(100–210)
	After baclofen	2300—(2200–2400)	10—(10–11)	1—(1–2)	200—(130–240)
2nd patient	Before baclofen	1900—(1850–1950)	15—(14–15)	3—(2–3)	120—(90–160)
	After baclofen	1850—(1800–1900)	11—(10–11)	1—(1–2)	160—(100–210)
3rd patient	Before baclofen	1950 (1850–2000)	14—(13–16)	3—(2–3)	140—(100–200)
	After baclofen	1900 (1800–2000)	10—(9–12)	1—(1–2)	200—(140–250)

Regarding urodynamic parameters, BC and RV increased, Pdetmax decreased, PVR was the same and DLPP was slightly decreased. Data on urodynamics are summarized in Table 2.

**Table 2 biomedicines-10-03266-t002:** Urodynamic data before and after ITB.

Urodynamic Data						
		BC (mL)	RV (mL)	Pdetmax (cmH_2_O)	PVR (mL)	DLPP (cmH_2_O)
1st patient	Before baclofen	187	123	77	12	77
	After baclofen	223	147	52	17	No leak
2nd patient	Before baclofen	152	104	81	0	81
	After baclofen	196	141	67	0	67
3nd patient	Before baclofen	148	136	62	13	62
	After baclofen	224	224	50	0	No leak

BC: bladder capacity, RV: reflex volume, Pdetmax: maximum detrusor pressure, PVR: post void residual, DLPP: detrusor leak point pressure.

## Data Availability

The data presented in this study are available on request from the corresponding author.

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
