# Peer review of "Intrathecal Baclofen for Spasticity: Is There an Effect on Bladder Function? Report of Three Cases and Review of the Literature"

_biomedicines, 2022, doi:10.3390/biomedicines10123266_

Round 1
Reviewer 1 Report
The article "Intrathecal baclofen for spasticity: Is there an effect on bladder function? Report of three cases and review of the literature "deals with an interesting subject. The Introduction and Material and Methos are clearly written. The results section is simple and sound, and the conclusion is relevant to the subject.
However, the study's limitations need to be clearly stated in a separate section since the authors presented only three cases. The same goes for the conclusions for intrathecal administration of baclofen advantage vs. baclofen oral administration. The same was stated in a sentence, "prospective studies are required to investigate the effect of ITB on bladder function "in abstract
The review of the literature is correct and relevant to the subject.
Author Response
Comment 1
The article "Intrathecal baclofen for spasticity: Is there an effect on bladder function? Report of three cases and review of the literature "deals with an interesting subject. The Introduction and Material and Methods are clearly written. The results section is simple and sound, and the conclusion is relevant to the subject.
Answer: Thank you for your kind words and the effort to review our manuscript
Comment 2
However, the study's limitations need to be clearly stated in a separate section since the authors presented only three cases.
Answer: We added the following paragraph at the end of the discussion
“The current study leads to results that cannot be generalized or applied in everyday practice. The main reasons are the small sample size and the fact that it explains an observation of change in bladder behavior as a positive side effect of therapy used to treat other symptoms. Furthermore, it concerns self-reported data that, objective as the measurements are, the assumptions cannot be independently verified. Therefore, prospective studies are required to investigate the effect of ITB on bladder function in the clinical field and at the basic science level.”
Comment 3
The same goes for the conclusions for intrathecal administration of baclofen advantage vs. baclofen oral administration.
Answer: We have described in details the advantages of Intrathecal baclofen administration comparing to oral intake in the 2nd paragraph of the introduction.
Comment 4
The same was stated in a sentence, "prospective studies are required to investigate the effect of ITB on bladder function "in abstract
Answer: We added this statement at the end of the discussion.
Comment 5
The review of the literature is correct and relevant to the subject.
Answer: Thank you for your effort to review our manuscript
Reviewer 2 Report
Dear Authors, I congratulat with your work. the topic os of main interest. The article is well written. There are some issues that may require clarification, in my opinion.
The restrospective nature of the report and the small number of patients (n=3) is a main limitation of the study. Although the size is very small, there is no statistical analysys. Is the reduction of pDet statistically significant? Is it clinically relevant? The apper lack a description of the population (age, BMI, comorbidities, therapy...).
I would state that TBI "may" result in DO: sometimes pseudo-dussynergia and detrusor underactivity may also arise, although DO is the prevalent dysfunction.
What do you mean with reflex volume? Was the evaluation of pDet dine in the filling or voiding phase?
The discussion could be shortened and include more practical applications of the results of the clinical experience.
Author Response
Comment 1
Dear Authors, I congratulate with your work. The topic is of main interest. The article is well written. There are some issues that may require clarification, in my opinion.
Answer: Thank you for your kind words and your effort to review our manuscript
Comment 2
The retrospective nature of the report and the small number of patients (n=3) is a main limitation of the study. Although the size is very small, there is no statistical analysis. Is the reduction of pDet statistically significant? Is it clinically relevant? The paper lack a description of the population (age, BMI, comorbidities, therapy...).
Answer: To describe the study limitations we added a separate paragraph at the end of the discussion. Although a statistical interpretation could be applied we believe that such a limited patients’ number cannot lead to safe conclusions even a statistical significant outcome could obtained. In the results we stated the clinical improvement of bladder function with the sentence: “Bladder function was improved in all patients. Regarding bladder diaries, incontinence episodes and micturitions were decreased, and the volume per void was slightly increased although the fluid intake was approximately the same”
With regard to the comment about the description of the population, the patients were all young, aged 36, 29 and 27 as mentioned in the first paragraph of the Material and Methods (line 68) and under treatment with “the same antimuscarinic medication (solifenacin succinate, 10 mg daily)” as mentioned in the midle of the 3rd paragraph of Material and Methods (lines 90-91). The patient aged 36 was also under treatment with statins but it was judged as irrelevant.
Comment 3
I would state that TBI "may" result in DO: sometimes pseudo-dussynergia and detrusor underactivity may also arise, although DO is the prevalent dysfunction.
Answer: we totally agree with your statement as in the 4th paragraph of the Introduction (line 58) we mention “…in the majority of cases” and in line 142 “such as NDO.” However, we added at this point (in line 57) that “TBI results predominantly in neurogenic overactivity amongst others (pseudodussynergia and detrusor underactivity)”.
Comment 4
What do you mean with reflex volume? Was the evaluation of pDet dine in the filling or voiding phase?
Answer: Reflex Volume is the volume of the first (involuntary) bladder contraction during the filling phase. The Pdetmax refers to the filing phase as it is described in the middle of the 3rd paragraph in the Materials and methods.
Comment 5
The discussion could be shortened and include more practical applications of the results of the clinical experience.
Answer: To address your comment, we added in the 1st paragraph of the discussion the sentence “The quality of life was improved as there were fewer incontinence problems and the overnight sleep hours were augmented. Maybe it is worth bearing in mind that selecting patients for ITB therapy should depend (apart from the local experience, the severe spasticity lasting at least six months and having failed to respond to the maximum recommended doses of oral antispasmodic medications) on whether it could result in a concomitant benefit on bladder function in addition to other therapies. It is strongly suggested that patients should be screened for response to the drug before implantation of the delivery pump.”
Round 2
Reviewer 2 Report
Dear Authors,
Thank you for having replied to my revision. The paper Is interesting, with the main limitations being represented by the retrospective nature and the small size of the cohort.